# Analysis of the Economic Burden of COVID-19 on the Workers of a Teaching Hospital in the Centre of Italy: Changes in Productivity Loss and Healthcare Costs Pre and Post Vaccination Campaign

**DOI:** 10.3390/vaccines11121791

**Published:** 2023-11-30

**Authors:** Sara Di Fabio, Giuseppe La Torre

**Affiliations:** Department of Public Health and Infectious Diseases, Sapienza University of Rome, 00185 Rome, Italy

**Keywords:** economic burden, COVID-19, healthcare workers, Italy, productivity loss, healthcare costs, vaccination

## Abstract

**Introduction:** Following the concerning levels of spread and severity of the infection, on 11 March 2020, the World Health Organisation declared the COVID-19 outbreak a pandemic. In response to the pandemic, governments adopted several mitigation strategies. The pandemic posed a great threat to the Italian healthcare workforce (HW), with Italy being one of the hardest-hit countries. The aim of this study is to estimate the economic burden of COVID-19 on the workforce of a teaching hospital in Central Italy. Two periods are compared: 1 March 2020–9 February 2021 vs. 10 February 2021–31 March 2022. **Methods:** This study is conducted from a societal perspective. The database (*n* = 3298) of COVID-19-confirmed cases among the HW was obtained from the occupational health office of the hospital. The first entry on the database refers to 1 March 2020. Cost data were used to assess the economic burden of COVID-19 on the hospital workforce. They include two main groups: hourly salaries and medical expenses. The cost of productivity loss, hospital admission, at-home treatments, and contact tracing and screening tests were computed for the first and second periods of the analysis. **Results:** The total economic burden during the first period is estimated to be around EUR 3.8 million and in the second period EUR 4 million. However, the average cost per person is smaller in the second period (EUR 1561.78) compared to the first one (EUR 5906). In both periods, the cost of productivity loss is the largest component of the economic burden (55% and 57%). The cost of hospital admission decreased by more than 60% in the second period. **Conclusion:** Outcomes of the analysis suggest that the economic burden of COVID-19 on the HW is higher in the first period of analysis compared to the second period. The main reason could be identified in the shift from more severe and critical confirmed cases to more asymptomatic, mild, and moderate cases. The causes of this shift are not easily detectable. Vaccination, variants of the virus, and an individual’s determinants of health could all be causes of the decrease in the economic burden of COVID-19 on the HW. COVID-19 can generate a high economic burden on healthcare workers and, more generally, on HWs. However, a comprehensive estimate of the economic burden of the pandemic needs to integrate the mental health repercussions and the long-term COVID-19 that will become evident in the coming years.

## 1. Introduction

### 1.1. Studying the Socio-Economic Burden of COVID-19

From the beginning of the COVID-19 pandemic in 2020, several studies were conducted on the socio-economic burden of the disease on the population. Most of the studies focused on the Disability Adjusted Life Years (DALYs) estimation for a country, reflecting the high number of SARS-CoV-2-related deaths [1,2,3,4,5,6,7]; however, fewer studies included the analysis of the burden of the disease in economic terms because of the loss of productivity [8,9,10,11], and even fewer studies were focused on the healthcare workforce or hospital workers [12]. In this study, the burden of COVID-19 on hospital workers was estimated in economic terms by computing the health-related productivity loss, the cost of hospitalisation, the cost of treatments, and the cost of screening and contact tracing activities.

#### 1.1.1. Health-Related Productivity Loss

Notwithstanding the extraordinary work of the HW, their dedication, and sacrifices, there were some health-related productivity losses due to COVID-19 [12,13,14,15]. Productivity loss due to health problems can be referred to as health-related productivity loss in which the main drivers are as follows: i.Absenteeism: being absent from work;ii.Presenteeism: working while being sick;iii.Inability to do unpaid work due to illness [12].

Most of the COVID-19 productivity losses are associated with the absenteeism that was demanded when members of the HW contracted the virus or had to quarantine because of positive contact or were affected by psychological disorders [12,14,15,16]. The latter may also cause presenteeism productivity loss, which is when members of the HW continue to work but are facing disorders like anxiety, depression, insomnia, and post-traumatic stress [12,13,16], which affect their performance. The combined cost of absenteeism and presenteeism constitute the cost of productivity loss and, therefore, the cost of the burden of disease. The human capital approach was used to estimate the productivity loss due to COVID-19, adopting the framework described by Pearce et al. [17] and Nurchis et al. [8]. Two out of the four types of productivity loss calculated by Pearce et al. [17] were of interest for the present study: i.Temporary time off work: individuals taking time off after the diagnosis;ii.Premature mortality: years of life lost because of the diagnosis.

Permanent time off, individuals ceasing to work because of the diagnosis, and reduced hours at work, i.e., those continuing to work but fewer hours, were not estimated as more data would have been necessary to categorise the absences from work. 

#### 1.1.2. Hospitalisation and At-Home Treatments 

Overall, in Italy, confirmed cases experiencing mild, moderate, and severe respiratory failure were admitted to the hospital, either in non-intensive care, semi-intensive care, or intensive care units. Confirmed cases diagnosed with pneumonia but not experiencing acute respiratory failure and confirmed cases without pneumonia were treated at home and, if necessary, assisted by local health services [18]. Hospitalisation and at-home treatments led to two different costs for the patient and society. While in the first case the costs were incurred by the national health system, as the Italian healthcare system is mostly public, in the second case there was an out-of-pocket expenditure for the patient in need of medicinal products. 

#### 1.1.3. Contact Tracing and Screening

To prevent the spreading of the disease among the HW, hospitals performed contact tracing and screening activities. At the teaching hospital that is the object of the present study, the contact tracing activity consisted of testing potential cases with individual molecular tests; potential cases were identified as those individuals who had been in contact with a confirmed positive case. The screening activity, on the other hand, consisted of testing hospital workers in pools at random every day. The pooling technique was implemented to reduce the costs of testing; however, it is cost-effective only when the probability of not finding positive cases is high. 

### 1.2. Perspective of the Study

This study was conducted from a societal perspective, including the hospital perspective. In Italy, the cost of productivity loss falls on the society rather than the single hospital because the Italian system dictates that the payment of the sick leave of workers is covered by national agencies. Indeed, the first day of absence from work of the worker who tested positive for COVID-19 was regarded as an injury and covered by the National Institute for Insurance against Occupational Accident (Istituto Nazionale per l’assicurazione contro gli infortuni sul lavoro–INAIL); then, once the worker was granted sick leave, the absence from work was covered by the National Institute for Social Security (Istituto nazionale della previdenza sociale–INPS). 

### 1.3. Objective of the Study

The aim of this study is to estimate the burden of disease and the related costs on the HW before and after the beginning of the vaccination campaign for COVID-19 in Italy. Therefore, two periods are compared: First period: from the first registered case among the hospital workforce on 1 March 2020 until 9 February 2021.Second period: from 10 February 2021 until 31 March 2022, the day when the Italian state of emergency ended.

The study was carried out in a 1200-bed hospital in Rome.

For the cut-off point between the two periods, the time needed to reach immunisation was considered and not the beginning of the vaccination campaign, 28 December 2020 [19]. Indeed, considering the vaccine with the longest timeline, immunisation is achieved after two weeks from the second dose, which is administered 21 days after the first dose; therefore, if a person was vaccinated with the first dose on 28 December 2020, the second dose would have been administered on 17 January 2021, and the immunisation would have been reached by 9 February 2021 [20,21]. 

## 2. Materials and Methods

### 2.1. Study Data

#### 2.1.1. Individual Data

Information about COVID-19-related absences was collected by the hospital’s occupational health office, which oversaw the contact tracing and screening activities within the hospital. The first entry on the database refers to 1 March 2020, when the first positive case of COVID-19 among the hospital’s workforce was recorded. 

The original database (*n* = 3298) used to perform the analysis included personal data (gender, birth date, role, and integrated care department of affiliation), date of the first positive test, date and results of subsequent COVID-19 tests, date of return to work, and information about the origin of the infection, recovery, and hospitalisation. Another dataset provided information about the symptoms experienced by confirmed cases in the first and second periods. The data were received already in an anonymised format and each entry was assigned an ID code. Furthermore, 99% of the entries were included in the analysis, while less than 1% (*n* = 22) were excluded due to the missing date of the first positive test. Age was calculated from the birth date, and the cases were assigned to four age clusters considering indications about the working age and the retirement age of the Italian population (18–31, 32–44, 45–57, 58–70). Moreover, to make the data uniform, roles and the integrated care departments of affiliation were standardised by reviewing the information available on the hospital’s website and comparing the entries in the dataset (Appendix A).

#### 2.1.2. Cost Data

Cost data were used to assess the cost of the burden of COVID-19 on the hospital workforce. They include two main groups: i.Hourly salaries were used to estimate the productivity loss with a human capital approach;ii.Medical expenses were used to estimate the cost of the disease for the society. The medical expenses refer to the cost of hospitalisation, the cost of treatments when not hospitalised, and the hospital cost of performing screening and contact tracing tests.

Hourly salaries for each category of hospital worker were retrieved from three sources:Gross hourly salary from the national contracts of hospital workers;Net hourly salary from previous micro-costing analysis conducted within the hospital, derived from interviews with hospital workers;Online job portals and reports from recruiting companies.

Overall, around 70% of the salaries came from national contracts, whereas the remaining 30% came from previous studies (8%), online job portals (17%), and interviews (5%) (Appendix A).

Data concerning medical expenses were obtained through interviews with hospital workers. In particular, the cost of hospitalisation and the cost of performing screening and contact tracing tests were obtained from administrative personnel, whereas the cost of treatments was obtained from general practitioners. The latter were also compared to online repositories of medicinal product costs. 

### 2.2. Methods

To allow the comparison between the two periods, all the hospital workers who tested positive for the first time before or on 9 February 2021 are considered in the first period (n_1_ = 654), whereas those who tested positive for the first time on or after 10 February 2021 are considered in the second period (n_2_ = 2621). The analyses of productivity loss, estimations of the cost of hospitalisation, and cost of treatment were performed on a sub-sample of entries, which were cleaned for missing values (Table 1).

#### 2.2.1. Estimation of the Cost of Productivity Loss

Hospital productivity loss—or costs due to absence from work—can be temporary, (TPL) when they are due to absenteeism, or permanent (PPL), when they are due to mortality [12]. The productivity loss was estimated through the human capital approach (HCA), following the methodology adopted by Pearce et al. (2015) and Nurchis et al. (2020). Individual TPL was estimated considering the days of missed work and valuing them with the attainable income for those days [14]. The days of missed work were derived from the difference between the date of return to work and the date of the first positive test. For the entries where the date of return to work was missing, a proxy was considered: one day after the date when the last negative result was registered, because we assume that individuals who received a negative result to a control test would return to work on the day of the negative result or the day after, and five days from when the last positive result was registered. Moreover, for medical directors, medical doctors, administrative personnel, Ph.D. students, residents, professors, researchers, and research fellows, two days were subtracted from the days of missed work for every seven days to account for weekends. 

National contracts of hospital workers provide salaries for 6 groups of hospital workers (Table 2); within each group, there are between 4 and 6 levels of salary according to the years of experience and expertise. As the dataset lacked information on the years of experience of each individual, an average of the salary levels per each group was attributed to the individuals with roles associated with the group (Table 3). The salaries of hospital workers employed through an external cooperative were considered to be 20% lower than the salaries of the workers with the same role but employed by the hospital. The percentage difference was estimated through interviews with hospital workers. 

To standardise the analysis, only gross values were considered; therefore, net salaries were increased by 33% to obtain the gross value. The transformation percentage is derived from an average of the differences of the available net and gross salaries for the same category of worker (*n* = 28). 

The individual TPL was computed considering the missed days of work for each individual and the daily gross salary attributed to each role (Formula (1)). The daily gross salary was computed considering the daily hours worked, which were derived from the weekly working hours included in the national contracts for hospital workers (38 h per week). The total TPL was obtained by summing individual TPLs.
(1)Individual TPL=Daily average salary per role×Days of missed work

The individual PPL was estimated similarly to the individual TPL, i.e., by multiplying the salary for the productive years of life lost (Formula (2)). The productive years of life lost were derived from the difference between the retirement age and the expected life expectancy at the age of the individual obtained from ISTAT Italian life tables [22]. The annual median salary was assumed to remain constant and correspond to the salary at the time of death.
(2)Individual PPL=Yearly average salary×Productive years of life lost

#### 2.2.2. Estimation of the Cost of Hospital Admission

The cost of hospital admission was computed for individuals for which there was an indication of being admitted to the hospital after resulting positive for SARS-CoV-2 (n_1_ = 38, n_2_ = 11). The individual LoS was computed through the difference between the date of discharge and the date of admission, which were extrapolated from the notes section of the dataset. For the 19 workers who were admitted to the hospital but did not present records of admission and discharge dates, a proxy of 12.59 days was used as their LoS [23]. The cost of hospital stay differs based on the type of hospitalisation: whether is an ordinary bed or an intensive care unit bed with or without automatic ventilation (Table 4). From the available data, only two individuals were admitted to intensive care units with automatic ventilation, all the others were admitted to an ordinary bed. The individual cost of hospital stay was obtained by multiplying the individual LoS by the cost of their type of hospital admission. The total cost was derived from the sum of individual costs.
(3)Individual Cost of hospitalisation=LoS×Cost of type of hospital admission

#### 2.2.3. Estimation of the Distribution of Confirmed Cases per Health State 

Confirmed cases were assigned to health states according to the number of symptoms experienced and their need for hospitalisation. The descriptions of the health states provided by the Istituto Superiore di Sanità [19] and the Italian national guidelines for the setting of assistance provided by AGENAS [18] were used to define the clinical manifestation associated with each health state (Table 5). Cases with zero symptoms were considered asymptomatic, cases experiencing one or two symptoms were mild, cases experiencing between three and five symptoms were moderate, cases experiencing between six and eight symptoms were severe, and cases with nine or more symptoms were critical.

#### 2.2.4. Estimation of the Cost of At-Home Treatment Options

The out-of-pocket cost of the at-home treatment options was estimated using the confirmed cases in the moderate health state (n_1_ = 144, n_2_ = 237). In both periods, it was assumed that general practitioners prescribed the treatments only to confirmed cases in the moderate health state, that is, cases experiencing between three and five symptoms and in no need of hospitalisation or just hospitalised and for whom local services were activated, e.g., the general practitioner prescribing the treatment. Two main treatment options can be distinguished. The first option includes the prescription of Zitromax and Deltacortene, which were administered by general practitioners to patients in need who tested positive for SARS-CoV-2 between March 2020 and December 2021. The second treatment option includes Eparina, Deltacortene, Tachipina, or Brufen, and it was administered between December 2021 and March 2022 (Table 6). For the sake of this analysis, it was assumed that the first treatment option corresponds to the first period (1 March 2020–9 February 2021) and the second treatment option to the second period (10 February 2021–31 March 2022). The total cost of the treatment during the first period was EUR 4.29 and during the second period EUR 1.29. The total cost of at-home treatments in the first and second periods was estimated by multiplying the number of confirmed cases in the moderate health state by the cost of the treatment. 

#### 2.2.5. Estimation of the Cost of Contact Tracing and Screening Activities

Contact tracing and screening activities were performed at the hospital level to monitor the spreading of the infection within the hospital, as well as prevent further spreading and protect hospital workers and patients. The average frequencies and costs of the tests performed by the hospital were provided by the administration (Table 7). 

For the period between 1 March 2020 and 9 February 2021, a total of 281 days was estimated as the days in which tests were performed, and for the period 10 February 2021–31 March 2022, it was a total of 350 days, excluding holidays and weekends. The total cost per day was computed by multiplying the cost per unit by the number of tests performed each day. The sum of the daily costs was then multiplied by the number of days to obtain the total cost for contact tracing and screening tests in the two periods.

## 3. Results

### 3.1. Distribution of Confirmed Cases and the Health States Model 

During the period March 2020–February 2021, there were 653 confirmed cases for SARS-CoV-2 among the HW; this number more than quadrupled in the second period (February 2021–March 2022), reaching 2621 confirmed cases. The daily distribution of confirmed cases among the HW shows that the peak of daily cases was registered in the second period, in January 2022 (Figure 1). On average, four members of the HW tested positive for SARS-CoV-2 daily in the first period and sixteen members of the HW in the second period. 

The age pyramids for the first and second periods show the distribution of the confirmed cases by age cluster and gender. Information on gender is available for 97% of the confirmed cases in the sample. The majority of female confirmed cases in the first period were in the age group 45–57 (33%), while the age group 18–31 had the majority in the second period (37%). Male confirmed cases experienced a similar distribution with respect to females, with 29% of male cases in the first period being aged 45–57 and 34% of male cases in the second period being aged 18–31 (Figure 2). Across periods, both females and males are shifting from a high number of cases in the older age clusters to a higher number of cases in younger ages (Figure 2).

Overall, more confirmed cases are recorded among females than males (61% in the first period and 65% in the second period). The total distribution of cases across age clusters is quite different between the two periods; at the beginning, the highest percentage of confirmed cases (31%) appears to be in the age cluster 45–57, while in the second period, the age cluster 18–31 seems the most affected one (34.9%) (Table 8). Only one death was registered during the first period for a male individual in the age cluster 45–57 (Table 8). 

Moving from the first to the second period, there is a shift from more moderate and severe cases to more asymptomatic and mild cases. Indeed, in the first period, the majority of the confirmed cases (37.6%) appear to be in the moderate health state group, that is, experiencing between three and five symptoms but not in need of hospitalisation or just admitted to the hospital. In the second period, as the percentage of moderate confirmed cases decreases to 18.4% and the percentage of severe cases reaches 1.3%, more cases are found in both the mild (46%) and asymptomatic (34%) health states (Figure 3).

Considering age clusters, among confirmed cases between 45–57 years of age, in the first period, 5% were in the critical health state and only 17% were asymptomatic, whereas in the second period, the same age group does not account for any critical case and more than 50% of the cases aged 45–57 experienced a mild health state. Overall, across all age clusters, the percentage of asymptomatic and mild cases increased from the first to the second period, and the percentage of moderate, severe, and critical cases decreased (Figure 4). 

A COVID-19 health states model was elaborated starting from the flow proposed by Wyper et al. [16] to identify the path of individuals undergoing testing for SARS-CoV-2 and the components of productivity loss due to positivity (PL_p_), productivity loss due to death (PL_d_), years of life lost (YLL), and years of life lost due to disability (YLD) (Figure 5). YLL and YLD estimate the Disability Adjusted Life Years (DALYs); however, they were not computed because this study only focused on the economic burden of COVID-19. 

The model mainly identifies two groups of individuals: i.Those with a positive test result, i.e., confirmed cases;ii.Those with a negative test result who have been in contact with a person testing positive for the infection, i.e., potential cases;

Within each group, two states were recognised: asymptomatic and symptomatic. The symptomatic state for those in the first group has four levels of severity (mild, moderate, severe, and critical) and could lead to death. For the symptomatic who tested negative, only the mild level is considered (Figure 5).

### 3.2. Productivity Losses 

The productivity losses were estimated on a sample of 598 confirmed cases for the first period and 1709 confirmed cases for the second period (Table 1), overlooking the entries without enough information on the days of absence from work. The average individual TPL for a confirmed case in the first period corresponds to EUR 3545 and on average 28 days of absenteeism and EUR 1374 and on average 13 days of absenteeism in the second period (Table 9). For the second period, the TPL is equivalent to the total PL, EUR 2786,336 (EUR 1374 per person); however, in the first period, it is necessary to account for the single case of death that occurred. Therefore, the total PL in the first period is EUR 2,120,196 (EUR 3545 per person), where EUR 317,329 is associated with mortality (Table 9). Generally, 63% of the total PL in the first period and 64% in the second period are attributed to female cases and, respectively, 37% and 36% to males.

Nevertheless, the individual TPL of females is higher than males in the first period (EUR 3152 for a female case, EUR 2809 for a male case) but not in the second period, when they are almost the same (EUR 1374 for a female, EUR 1375 for a male) (Table 10). Across age clusters, the highest PL is recorded for the fourth age cluster (58–70), and it is very similar for females and males across the periods: in the first period, EUR 3446 for females aged and EUR 3447 for males; in the second period, EUR 1814 for females and EUR 1782 for males (Table 10). Females in the age cluster 45–57 contributed the most to the total TPL, with 23% of the total in the first period and 19% in the second period. Males contributed overall less than females to the PL, with the lowest contribution being for males in the age cluster 18–31 in the first period (5%). The contributions to the total TPL are aligned to the average number of days of missed work. Indeed, in the first period, females in the age cluster 45–57 have the highest average of days of missed work (34.7), and males in the age clusters 18–31 and 32–44 have the lowest average of days of missed work (22.2). In the second period, the average days of missed work are significantly lower than in the first period. Females in the age cluster 58–70 on average missed the most workdays (16.9 days), whereas females in the age cluster 18–31 missed the least workdays on average (11.1 days) (Table 10). 

### 3.3. Hospital Admissions

Overall, 38 confirmed cases were admitted to a hospital during the first period and 11 in the second period. All the cases recovered except for one death. The total cost of hospitalisation largely decreased from the first to the second period, from EUR 463,440 to EUR 151,770. The total LoS is higher in the first period than in the second period, respectively, 476 days and 151 days, but the average LoS was 13 in the first period and 14 in the second period. The highest number of hospital admissions was recorded for the age cluster 45–57 in both periods, 20 individuals, and it was higher for females than males, while the lowest number of hospital admissions was recorded for the age cluster 18–31, 3 individuals. Although females were admitted more often to the hospital (in the first period 23 F and 14 M, in the second period 7 F and 4 M), the average LoS is slightly higher for males than females: in the first period, 12 days for females and 14 days for males, and in the second period, respectively, 12 and 18 days (Table 11). The average cost of confirmed cases admitted to a hospital in the first period is slightly lower than the average cost for the second period, respectively, EUR 12,196 and EUR 13,797 (Table 11). Overall, female cases generated higher costs of hospital admission than male cases in both periods. The most expensive age cluster for females was the one with the largest number of confirmed cases admitted to the hospital (45–57), EUR 139,180 in the first period and EUR 60,180.00 in the second period (Table 11).

### 3.4. At-Home Treatments 

The estimation of the cost of at-home treatments for confirmed cases not admitted to a hospital was computed for cases in the moderate health state in the first and second periods. The number of moderate cases was computed from the hospital’s database; a total of 144 moderate cases in the first period and 237 in the second period were identified. Therefore, the total cost of at-home treatments for confirmed cases in the moderate health state was EUR 618 in the first period and EUR 306 in the second period, registering a decrease of about 50% between the two periods (Table 12).

### 3.5. Contact Tracing and Screening Tests 

Part of the hospital’s expenditure in monitoring and preventing the spread of the infection within the HW was estimated with the cost of contact tracing and the screening tests performed. The total expenditure incurred by the hospital was EUR 2,871,050 for 473,250 tests performed. Most of the performed tests were pooled, meaning that samples from 100 individuals were tested together for the infection (140,500 during the first period and 175,000 during the second period) (Table 13).

### 3.6. Total Economic Burden of COVID-19 on the HW

Comparing the two periods of analysis, the total economic burden during the second period is higher than in the first period, respectively, EUR 4,093,417 and EUR 3,862,803. However, considering the number of confirmed cases in both periods, the cost per individual in the second period is significantly lower than the cost per individual in the first period: EUR 1562 in the second period and EUR 5906 in the first period (Table 14). The composition of the total economic burden on the HW is very similar between the two periods, with most of the costs attributed to the PL (55% in the first period and 57% in the second period) and a very small percentage corresponding to at-home treatments, less than 1%. The cost composition shows a decrease in the cost of hospital admission from 12% in the first period to 4% in the second period (Figure 6).

## 4. Discussion

COVID-19 affected every healthcare system, generating growing concerns about the health of healthcare workforces because of both mortality and morbidity. Health-related productivity losses and cost of treatments for confirmed cases measure how much the system was affected and provide insights into how to better plan for the future and increase the sustainability and resiliency of the healthcare system. One of the main limitations faced by healthcare systems was to effectively record mortality and morbidity due to COVID-19, as untested individuals may not be included in the national database about COVID-19 deaths and morbidity due to the pandemic still needing to be completely defined. In particular, long-term COVID-19 and mental health concerns that will affect future presenteeism productivity loss should be regarded. 

The dataset used in this study included 3298 confirmed cases recorded among the HW of a teaching hospital located in Central Italy, of which 654 cases tested positive for SARS-CoV-2 in the first period of the analysis (1 March 2020–9 February 2021) and 2621 cases in the second period (10 February 2021–31 March 2022). The purpose of this study was to quantify the economic burden of COVID-19 on the HW by estimating the cost of productivity loss due to absenteeism, the cost of hospital admissions, the cost of at-home treatment, and the cost of contract tracing and screening activities.

The diagnostic options were fully available in the hospital we studied, as we acted as HUB for the performance of diagnostic testing for SARS-CoV-2 in Rome. Moreover, concerning the availability of PPE, this hospital began to buy the devices and the PPE at the very beginning of the pandemic, due to a centralisation of the buying process at the regional level, so that the costs were mitigated, especially in the first waves of the pandemic.

Overall, the total burden of COVID-19 on the HW reflects a shift from severe and critical cases to more asymptomatic, mild, and moderate cases when moving from the first to the second period of analysis; indeed, although the cost of hospital admissions decreased by more than 60% in the second period, the cost of productivity loss increased by 11%, suggesting that more confirmed cases were absent from work but their health state was not as severe as in the first period, thus fewer cases required hospitalisation. In the first period, almost 6% (38 cases) of the sample was hospitalised, and it was less than 1% in the second period (11 cases). The cost of hospital stay did not change, as this is a value that is fixed at the national level due to DRGs. The cost of hospitalisation falls mostly on society because of the public nature of the Italian healthcare system. Hospitals are entitled to reimbursement for treated patients from the region in which they are located. The Diagnosis Related Group (DRG) for COVID-19 foresees an average reimbursement of EUR 9062 per patient requiring hospitalisation because of COVID-19. 

After the introduction of the vaccine in December 2020 until the discovery of Delta and Omicron, in autumn 2021, the number of daily hospitalisations among the HCW was stable and relatively low; however, with the new variants, a slight increase was registered. Studies show that vaccinated individuals are less likely to be hospitalised or to become severe and critical cases [24,25]. Since this analysis is focused on a very specific population with distinct characteristics, including the mandatory vaccination for COVID-19, it is arguable to assume that vaccination is one of the leading causes for the decrease in hospitalisation and hospital admissions that occurred moving from the first to the second period. However, vaccination is not the only cause for the decrease in costs, research on the variants of the virus [26,27,28], on how the virus is transmitted, and on the determinants of health [29,30,31,32] add an extra layer of complexity and prevent the identification of direct causes of infection and hospitalisation due to COVID-19. 

The total cost of productivity loss was estimated to be EUR 2.1 million in the first period and EUR 2.3 million in the second period, with an average cost per person of EUR 3545 in the first period and EUR 1374 in the second period. During the first period, there were 17,191 days of absenteeism in total, and in the second period there were 22,530; on average, there were 28.8 days per person in the first period and 13.2 days in the second period. Although the total figures for the second period are higher than for the first period, the cost of productivity loss due to absenteeism per person and the average days of missed work were lower compared to the first period, supporting the evidence of a shift in the distribution of health states. A similar study (*n* = 1958) conducted on the hospitals of Mashhad University of Medical Sciences in Iran [14] reports an average of 16 days of absenteeism per person, with a total cost due to absenteeism of nearly USD 1.24 million and an average cost of absenteeism of about EUR 640. Other studies estimated the cost of absenteeism on healthcare workers in the general population, instead of a specific hospital, and the general population of a country. Maltezou et al. [33] (*n* = 3398) analysed the direct and indirect costs of absenteeism and presenteeism of the HCW in Greece. Their results show that confirmed cases among the HCW (*n* = 252) were absent from work for an average of 25.8 days, and the cost of absenteeism due to COVID-19, computed with an HCA, totalled EUR 552,500. Nurchis et al. [8] analysed the productivity loss due to COVID-19 on the general Italian population, predicting around EUR 100 million to be the total cost of productivity loss due to absenteeism for all the working age classes and around EUR 300 million of productivity loss due to mortality for all the working age classes. 

Regarding at-home treatments, the costs represent the out-of-pocket expenditure of the individual but do not include the expenditure of the healthcare system. Indeed, because of the public nature of the Italian system, the cost of medicinal products falls mostly on the system, while the citizen to whom the medicinal product is prescribed has to pay a minimum fee, the “ticket”. Between the two periods, as more discovery and research about the infection and treatment options became available, the guidelines for administering at-home treatments evolved. Therefore, the difference between the two periods is correlated with a change in treatment resulting in different prices and a change in the number of moderate confirmed cases who were considered suitable subjects for this intervention. 

Looking at the distribution of confirmed cases among the HW, in the second period, the highest percentage of confirmed cases is recorded in the age cluster 18–31; this may be linked to the increased number of medical students and new graduates who were undergoing training or joined the hospital workforce. The Italian government introduced recruitment schemes for recent graduates to overcome the large shortage of healthcare workers that got worse after the first year of the pandemic [34]. 

The main strength of this study was the large sample size; however, some limitations were identified. Regarding contact tracing and screening costs, the increase between the two periods is only due to the number of days in the intervals and not to the number of tests performed, representing a limitation of the study. The costs of tests performed are to be considered together for the first and second periods and estimate part of the hospital’s expenditure in preventing the spreading of the infection and monitoring the cases within the HW population. Another limitation of the study was the lack of information on the HW who tested negative over the entire period of the analysis (March 2020–March 2022) and who took a leave from work not correlated with testing positive for COVID-19. Therefore, it was not possible to estimate the factors associated with absenteeism due to COVID-19. Moreover, we need to recognize that we were not able to retrieve the performance retribution and seniority of healthcare professionals. Lastly, the findings of the research are not generalisable to Italian HCWs even though similar methods of analysis could be implemented in other contexts.

## 5. Conclusions

The results of this study show that COVID-19 can generate a high economic burden on HCWs and, more generally, on the HW. The largest economic burden is represented by the cost of productivity loss due to absenteeism from work. Comparing the periods before and after the vaccination, there was a reduction in the average economic burden of COVID-19; however, this reduction is not entirely attributable to the vaccination campaign. Indeed, other causes for the reduction include preventive and protective measures implemented at the national and hospital level, the spreading of new variants of the virus, and an individual’s determinants of health.

Retrospective and observational analyses are needed to achieve a more comprehensive measure of the socio-economic burden of COVID-19 on the HW, considering mental health repercussions and long-term COVID-19 as well. Further analyses will need to overcome the current limitations, thus including data about the exact number and results of the tests performed on the HW either for contact tracing or screening. These data would allow the identification of the factors impacting the costs explored in this analysis.

The COVID-19 pandemic has been threatening healthcare systems and societies, changing lifestyles, and forcing populations to adapt to a “new normal” in their personal and professional lives. The socio-economic burden of the pandemic should also take into consideration the effects of the digital transformation. Technology has allowed for remote and smart working, increasing an individual’s productivity in some cases because of the higher autonomy and self-leadership, but at the same time increasing stress and distractions. In the healthcare industry, COVID-19 fast-tracked the implementation, or at least the discussion, of several innovations such as Telehealth and Telemedicine, remote monitoring, and decentralised clinical trials [35,36].

Findings from this study and related further studies could provide insights on the budget allocation for preventive and protective measures to be implemented at the hospital level that can have an impact on the costs associated with absenteeism and presentism by acting on the working conditions of the HW.

## Figures and Tables

**Figure 1 vaccines-11-01791-f001:**
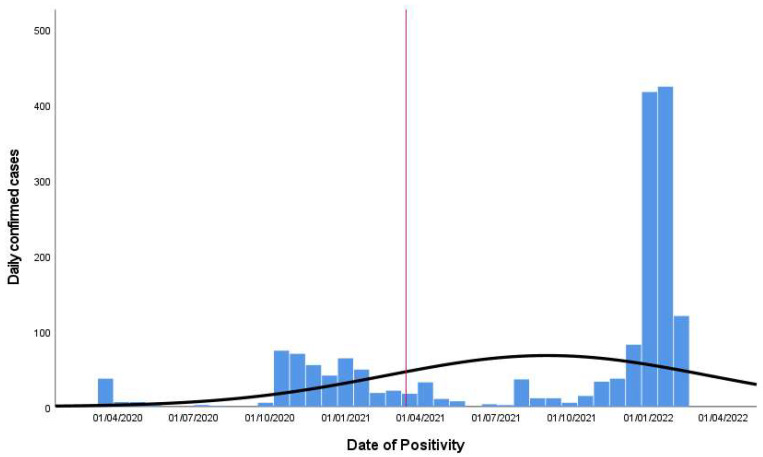
Frequency distribution of daily confirmed cases in the first and second periods. The red line corresponds to the passage between the two periods.

**Figure 2 vaccines-11-01791-f002:**
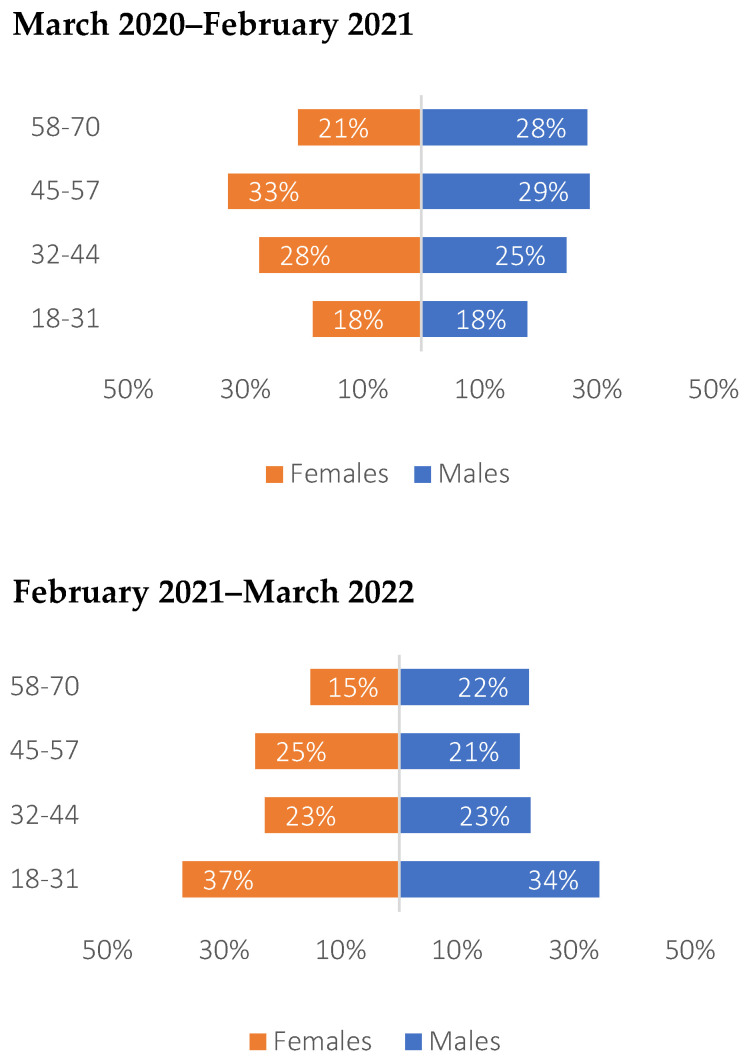
Confirmed cases of hospital workers included in the study by gender and age clusters in the first and second periods.

**Figure 3 vaccines-11-01791-f003:**
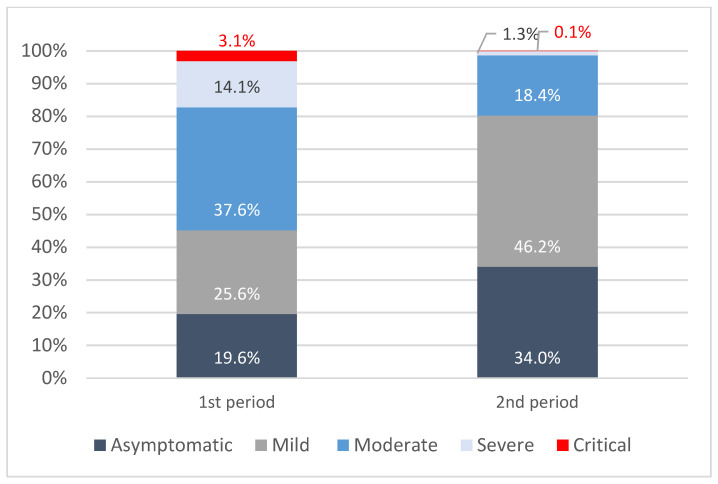
Distribution of confirmed cases per health state in the first and second periods.

**Figure 4 vaccines-11-01791-f004:**
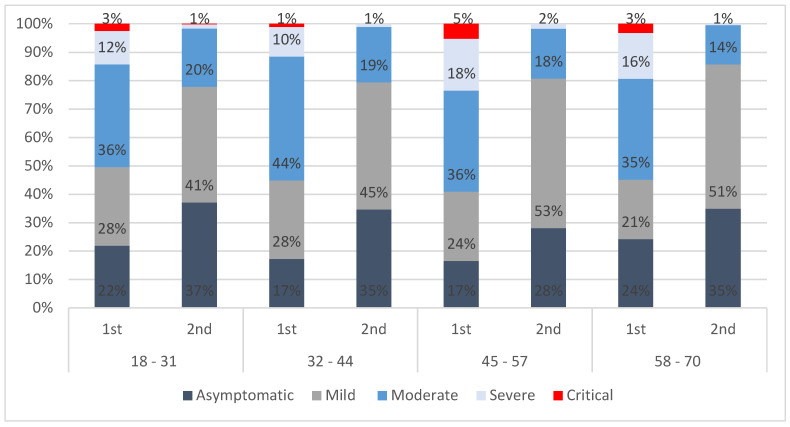
Percentage of confirmed cases per health state and age cluster in the first and second periods.

**Figure 5 vaccines-11-01791-f005:**
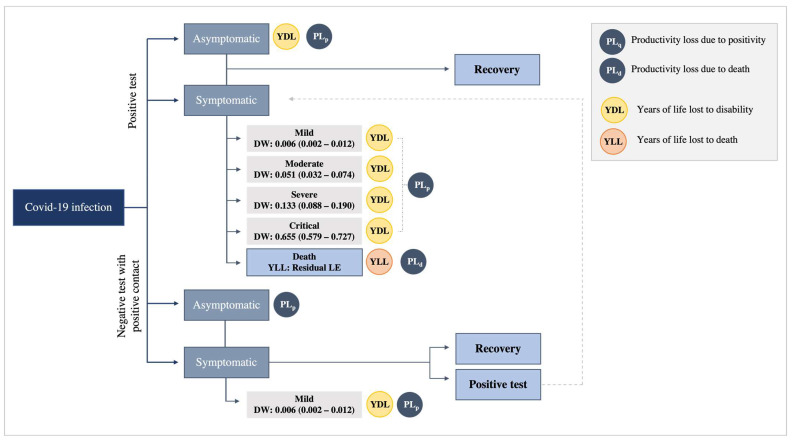
COVID-19 health states model. Source: Elaborated from Wyper et al. [16].

**Figure 6 vaccines-11-01791-f006:**
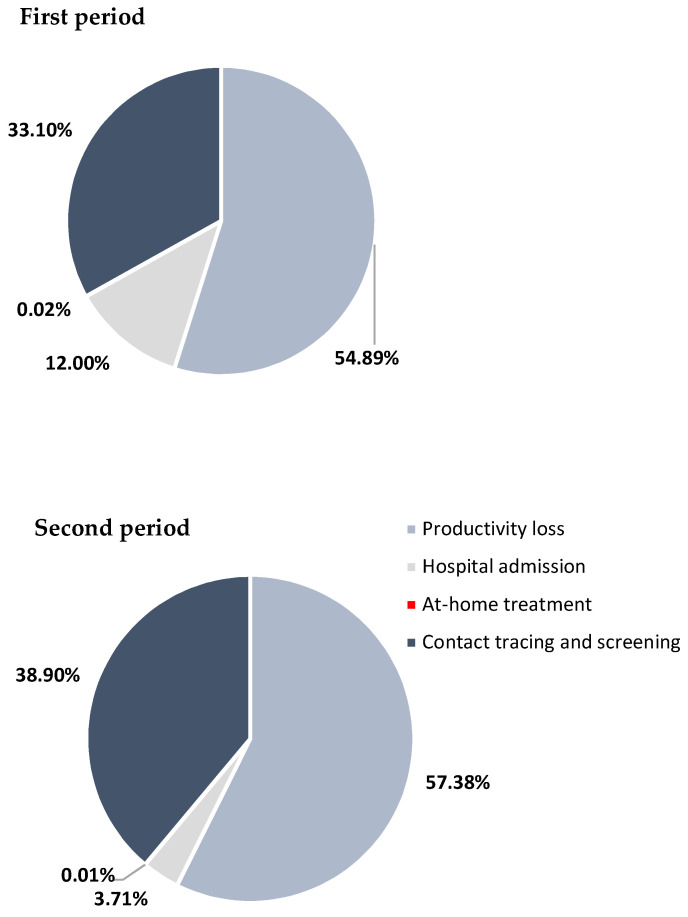
Composition of the total economic burden of COVID-19 on the HW in the first and second periods.

**Table 1 vaccines-11-01791-t001:** Samples for the different types of analyses during the first (n_1_) and second (n_2_) periods.

ANALYSIS	n1	n2	DESCRIPTION
TOTAL SAMPLES	654	2621	All registered confirmed cases
ESTIMATION OF THE COST OF PRODUCTIVITY LOSS	598	1709 ^1^	Confirmed cases for which the days of missed work were available
ESTIMATION OF THE COST OF HOSPITALISATION	38	11	Confirmed cases for which data on hospitalisation were available
ESTIMATION OF HEALTH STATES	380	1287	Confirmed cases for which data on experienced symptoms were available
ESTIMATION OF THE COST OF TREATMENT	144	237	Confirmed cases in the moderate health state

^1^ The nine entries of the n2 for the analysis of the productivity loss refer to April 2022; the last case was registered on 6 April 2022.

**Table 2 vaccines-11-01791-t002:** Monthly, daily, and hourly average salary per group of hospital workers.

GROUP	AVERAGE GROSS SALARY (EUR)		
	Monthly	Daily	Hourly
**A**	1523.78	58.61	9.77
**B**	1656.26	63.70	10.62
**C**	1706.85	65.65	10.94
**D**	1798.35	69.17	11.53
**DS**	1892.27	72.78	12.13
**E**	1845.31	70.97	11.83

**Table 3 vaccines-11-01791-t003:** Salary groups of hospital workers and associated roles.

**A**	o CLEANING ATTENDANT	o CANTEEN CLERK
	o Healthcare auxiliary	o Social welfare worker
	o Cloakroom attendant	
**B**	o Driver	o Childhood vigilante
	o Maintenance	o Teacher
	o Cook	o Fleet operator
	o Social health operator	o Parking contact person
	o Healthcare worker	
**C**	o Administrative	o ICT technician
	o Social child health operator	o Security
	o Specialised technical operator	o Reception service in DEA
	o Technical assistant	
**D**	o Nurse	o Speech therapist
	o Obstetrician	o Orthoptist and assistant of ophthalmology
	o Dietician	o Orthoptic director
	o Healthcare assistant	o Neuro and psychomotor therapist of the developmental age
	o Podiatrist	o Psychiatric rehabilitation technician
	o Dental hygienist	o Occupational therapist
	o Biomedical laboratory health technician	o Social worker
	o Radiologist technician	o Chaplain
	o Neurophysiopathology technician	o Administrative assistant
	o Orthopedic technician	o Technical manager
	o Cardiocirculatory physiopathology and cardiovascular perfusion technician	o Head of Technical Health Service of Medical Radiology (THSMR)
	o Dental technician	o Coordinator of Technical Health Service of Medical Radiology (THSMR)
	o Audiometrist technician	o Technician
	o Hearing aid technician	o Prevention technician in the environment and in the workplace
	o Physiotherapist	
**DS**	o Head nurse	o Nursing coordinator
	o Obstetric coordinator	o Technical coordinator
**E**	o Healthcare executive	o Medical director
	o Complex Operational Unit Director (COU)-Medical	o Departmental Area Manager
	o Complex Operational Unit Director (COU)-Not medical	

**Table 4 vaccines-11-01791-t004:** Cost of hospital stay based on the type of admission and the average cost of hospital stay per day.

TYPE OF HOSPITAL ADMISSION	COST PER DAY (EUR)
Ordinary hospital bed	1000.00
Cost of intensive care w/o automatic ventilation	1315.00
Cost of intensive care w automatic ventilation	1654.00
Average cost of hospital stay per day	1323.00

**Table 5 vaccines-11-01791-t005:** Description, clinical manifestation, and setting for assistance for each health state.

NAME	DESCRIPTION	CLINICAL MANIFESTATIONS	SETTING FOR ASSISTANCE
ASYMPTOMATIC	Positive test result for SARS-CoV-2 without symptoms or signs of the disease.	Absence of symptoms	Home and potentially activate local services
MILD	Has a low fever and mild discomfort but no difficulty with daily activities. Does not require hospitalisation.	Presence of 1 or 2 symptoms with no need of medical attention	Home without activation of local services
MODERATE	Has a fever and aches, and feels weak, which causes some difficulty with daily activities. Home quarantining or just admitted to hospital.	Presence of 3–5 symptoms, in need of medical attention but not hospitalised	Home with activation of local services
SEVERE	Has a high fever and pain and feels very weak, which causes great difficulty with daily activities. Requires hospitalisation.	Presence of 6–8 symptoms, in need of hospitalisation	Hospital, ordinary bed, or semi-intensive care unit
CRITICAL	Positive test result for SARS-CoV-2 with clear signs and symptoms of the disease that require intensive care unit admission (with or without respiratory support).	Presence of 9 or more symptoms, in need of intensive care unit	Hospital, intensive care unit

**Table 6 vaccines-11-01791-t006:** Cost and dosage of at-home treatments prescribed by general practitioners during the two periods of the analysis.

	MEDICINAL PRODUCT	DOSE	PRICE/UNIT	UNITS	TOTAL PRICE
**Treatment 1** **1 Mar 2020–9 Feb 2021**	Zitromax or generic	3 pills/day per 6 days	EUR 1.50	2	EUR 3.00
	Deltacortene	25 mg/day per 3 days and then reduced dose	EUR 1.29	1	EUR 1.29
		**Total price per treatment**			**EUR 4.29**
**Treatment 2** **10 Feb 2021–31 Mar 2022**	Eparina-Clexane	4000 units/day per 12–18 days	-	3	-
	Deltacortene	25 mg/day per 3 days and then reduced dose	EUR 1.29	1	EUR 1.29
	Tachipirina/Brufen	2 pills/day per 5 days	-	1	-
		**Total price per treatment**			**EUR 1.29**

**Table 7 vaccines-11-01791-t007:** Daily hospital expenses for different types of SARS-CoV-2 tests.

SARS-CoV-2 TEST	COST/UNIT	QUANTITY/DAY	TOTAL COST/DAY
Cost of molecular test for contact tracing	EUR 19.00	150	EUR 2850.00
Cost of molecular test for screening (in pooling)	EUR 2.00	500	EUR 1000.00
Cost of rapid test	EUR 7.00	100	EUR 700.00

**Table 8 vaccines-11-01791-t008:** (**a**) Confirmed cases and deaths per gender and age cluster during the first period (1 March 2020–9 February 2021)*;* (**b**) second period (10 February 2021–31 March 2022).

(a)												
AGE CLUSTER	Males				Females				Total Cases			
	Confirmed Cases	% of Total Cases	Deaths	% of Total Deaths	Confirmed Cases	% of Total Cases	Deaths	% of Total Deaths	Confirmed Cases	% of Cases per Age Cluster	Deaths	% of Death per Age Cluster
18–31	46	38.7%	0	-	73	61.3%	0	-	119	18.2%	0	-
32–44	63	36.6%	0	-	109	63.4%	0	-	172	26.3%	0	-
45–57	74	36.3%	1	100%	130	63.7%	0	-	204	31.2%	1	100%
58–70	72	46.5%	0	-	83	53.5%	0	-	155	23.7%	0	-
NOT AVAILABLE	2	66.7%	0	-	1	33.3%	0	-	3	0.5%	0	-
**TOTAL**	257	39.4%	1	100%	396	60.6%	0	-	653	100%	1	100%
(b)												
18–31	302	33.0%	0	-	613	67.0%	0	-	915	34.9%	0	-
32–44	198	34.3%	0	-	379	65.7%	0	-	577	22.0%	0	-
45–57	182	30.9%	0	-	407	69.1%	0	-	589	22.5%	0	-
58–70	196	43.9%	0	-	250	56.1%	0	-	446	17.0%	0	-
NOT AVAILABLE	42	44.7%	0	-	52	55.3%	0	-	94	3.6%	0	-
**TOTAL**	920	35.1%	0	-	1701	64.9%	0	-	2621	100%	0	-

**Table 9 vaccines-11-01791-t009:** Estimated temporary, permanent, and total productivity loss in the first and second periods and per person calculated with the HCA.

PERIOD	COST OF TEMPORARY PL	COST OF PERMANENT PL	COST OF TOTAL PL	AVERAGE TPL PER PERSON
**1 Mar 2020–9 Feb 2021**	EUR 1,802,866.04	EUR 317,329.68	EUR 2,120,195.72	EUR 3545.48
**10 Feb 2021–31 Mar 2022**	EUR 2,348,841.18	-	EUR 2,348,841.18	EUR 1374.40

**Table 10 vaccines-11-01791-t010:** Estimated temporary productivity loss in the first period and second period by gender and age cluster calculated with the HCA.

**1 March 2020–9 February 2021**							
**Row Labels**	**Count of Age Cluster**	**Sum of Days of Missed Work**	**Sum of Gross Individual TPL (EUR)**	**Avg. TPL per Age Cluster (EUR)**	**% of TPL per Gender and Age**	**Avg. Individual TPL (EUR)**	**Days of Missed Work per Person**
**F**	**366**	**11,121.6**	**1,153,810.78**		**64%**	**3152.49**	**30.4**
**18–31**	65	1541.1	139,855.78	2151.63	7.8%	2151.63	23.7
**32–44**	102	3103.4	329,608.21	3231.45	18.3%	3231.45	30.4
**45–57**	122	4229.7	418,980.78	3434.27	23.2%	3434.27	34.7
**58–70**	77	2247.3	265,366.01	3446.31	14.7%	3446.31	29.2
**M**	**231**	**6070.0**	**649,055.26**		**36.0%**	**2809.76**	**26.3**
**18–31**	41	909.9	90,808.25	2214.84	5.0%	2214.84	22.2
**32–44**	59	1309.4	125,728.71	2131.00	7.0%	2131.00	22.2
**45–57**	63	1922.3	198,063.43	3143.86	11.0%	3143.86	30.5
**58–70**	68	1928.4	234,454.87	3447.87	13.0%	3447.87	28.4
**GRAND TOTAL**	**597**	**17,191.6**	**1,802,866.04**		**100.0%**	**3019.88**	**28.8**
**10 February 2021–31 March 2022**							
**Row Labels**	**Count of Age Cluster**	**Sum of Days of Missed Work**	**Sum of Gross Individual TPL (EUR)**	**Avg. TPL per Age Cluster (EUR)**	**% of TPL per Gender and Age**	**Avg. Individual TPL (EUR)**	**Days of Missed Work per Person**
**F**	**1104**	**15,045.6**	**1,516,907.14**		**64.6%**	**1374.01**	**13.6**
**18–31**	343	3820.0	362,415.05	1056.60	15.4%	1056.60	11.1
**32–44**	284	3726.4	377,726.01	1330.02	16.1%	1330.02	13.1
**45–57**	296	4439.1	448,347.41	1514.69	19.1%	1514.69	15.0
**58–70**	181	3060.0	328,418.67	1814.47	14.0%	1814.47	16.9
**M**	**605**	**7485.3**	**831,934.04**		**35.4%**	**1375.10**	**12.4**
**18–31**	195	2081.3	210,780.09	1080.92	9.0%	1080.92	10.7
**32–44**	131	1550.9	167,317.60	1277.23	7.1%	1277.23	11.8
**45–57**	133	1852.0	193,608.15	1455.70	8.2%	1455.70	13.9
**58–70**	146	2001.1	260,228.20	1782.38	11.1%	1782.38	13.7
**GRAND TOTAL**	**1709**	**22,530.9**	**2348,841.18**		**100.0%**	**1374.40**	**13.2**

**Table 11 vaccines-11-01791-t011:** Cost of hospital admission by gender and age in the first and second periods.

**1 Mar 2020–9 Feb 2021**						
**Row Labels**	**Count of Age Cluster**	**Days of Hospital Stay**	**Sum of Cost of Hospital Stay (EUR)**	**Avg. Cost of Hospital Stay per Age Cluster (EUR)**	**Avg. Cost of Hospital Stay per Gender (EUR)**	**Avg. Tot Cost of Hospital Stay per Person (EUR)**
**F**	**23**	**12**	**275,900.00**		**11,995.65**	
**18–31**	2	13	25,180.00	12,590.00		
**32–44**	3	11	34,180.00	11,393.33		
**45–57**	10	14	139,180.00	13,918.00		
**58–70**	8	10	77,360.00	9670.00		
**M**	**14**	**13**	**187,540.00**		**13,395.71**	
**18–31**	-	-	-	-		
**32–44**	2	12	23,590.00	11,795.00		
**45–57**	4	16	63,770.00	15,942.50		
**58–70**	9	13	100,180.00	12,522.50		
**GRAND TOTAL**	**38**	**13**	**463,440.00**	**12,590.00**		**12,195.79**
**10 Feb 2021–31 Mar 2022**						
**Row Labels**	**Count of Age Cluster**	**Days of Hospital Stay**	**Sum of Cost of Hospital Stay (EUR)**	**Avg. Cost of Hospital Stay per Age Cluster (EUR)**	**Avg. Cost of Hospital Stay per Gender (EUR)**	**Avg. Tot Cost of Hospital Stay per Person (EUR)**
**F**	**7**	**12**	**82,180.00**		**11,740.00**	
**18–31**	-	-	-	-		
**32–44**	-	-	-	-		
**45–57**	6	10	60,180.00	10,030.00		
**58–70**	1	22	22,000.00	22,000.00		
**M**	**4**	**17**	**69,590.00**		**17,397.50**	
**18–31**	1	15	15,000.00	15,000.00		
**32–44**	-	-	-			
**45–57**	-	-	-	-		
**58–70**	3	18	54,590.00	18,196.67		
**GRAND TOTAL**	**11**	**14**	**151,770.00**			**13,797.27**

**Table 12 vaccines-11-01791-t012:** Cost of at-home treatments for confirmed cases in the moderate health state in the first and second periods.

	PERIOD 1	PERIOD 2
Cost of at-home treatment per person	EUR 4.29	EUR 1.29
Moderate confirmed cases	144	237
**Total cost of at-home treatment**	**EUR 617.76**	**EUR 305.73**

**Table 13 vaccines-11-01791-t013:** Cost of contact tracing and screening tests performed by the hospital on the hospital’s workforce.

	FIRST PERIOD		SECOND PERIOD		TOTAL	
Type of test	N. of Tests	Expenditure (EUR)	N. of Tests	Expenditure (EUR)	Total Tests	Total Expenditure (EUR)
**Cost of molecular test for contact tracing**	42,150	800,850	52,500	997,500	94,650	1,798,350
**Cost of molecular test for screening (in pooling)**	140,500	281,000	175,000	350,000	315,500	631,000
**Cost of rapid test**	28,100	196,700	35,000	245,000	63,100	441,700
**Total**	210,750	1,278,550	262,500	1,592,500	**473,250**	**2,871,050**

**Table 14 vaccines-11-01791-t014:** Cost of PL, hospital admission, at-home treatment, and contact tracing and screening in the first and second periods.

ESTIMATES	FIRST PERIOD	SECOND PERIOD
Productivity loss (EUR)	2,120,195.72	2,348,841.18
Hospital admission (EUR)	463,440.00	151,770.00
At-home treatment (EUR)	617.76	305.73
Contact tracing and screening (EUR)	1,278,550.00	1,592,500.00
**Total (EUR)**	**3,862,803.48**	**4,093,416.91**
Confirmed cases	654.00	2,621
**Total per person (EUR)**	**5906.43**	**1561.78**

## Data Availability

Data are available asking the authors.

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
