# Peer review of "Analysis of the Economic Burden of COVID-19 on the Workers of a Teaching Hospital in the Centre of Italy: Changes in Productivity Loss and Healthcare Costs Pre and Post Vaccination Campaign"

_vaccines, 2023, doi:10.3390/vaccines11121791_

Round 1

Reviewer 1 Report

Comments and Suggestions for Authors

Estimated Authors,

Estimated Editors,

I've been asked to provide a peer review report on this cost-analysis report from Latorre and Di Fabio.

Authors, through the retrospective analysis of data from a teaching hospital from Central Italy, provided a well detailed and well performed analysis of direct and direct costs of SARS-CoV-2 pandemic on the total hospital workforce.

The study would deserve a full publication on high-quality Q1-Q2 journals, but I've some concerns/doubts that should be addressed before a potential acceptance.

First of all, Authors should provide some further details on the teaching hospital that provided included data. While I understand that partial anonymization of data is instrumental in coping with EU GDPR and Ethical issues, some further information about the size of the hospital, the total number of beds/admissions before and during the pandemic, are required to guarantee a better characterization of the settings. Some epidemiological details about the occurrence of SARS-CoV-2 pandemic is moreover highly in need: Italy has been severely affected by the pandemic since its inception, but the Italian pandemic was quite heterogeneous in the early stages, with high rates in Lombardy, Emilia Romagna, Veneto and Piedmont, but moderate or even low rates in central and southern regions. This potential issue should be discussed extensively in the later sections, as the baseline epidemiology of SARS-CoV-2 has reasonably affected the eventual estimates. 

Second, some issues are elicited by Salary group E. According to your Table 3, Group E includes medical professionals with a very high but also heterogeneous qualification. In fact, the AVERAGE monthly income is lower than that of group DS. Moreover, the monthly income is strictly associated with seniority of medical professionals, and is increased (for medical professionals) by additional retribution for performance. As medical professionals have been substantially affected by SARS-CoV-2 pandemic, particularly in early stages, these shortcomings have reasonably affected your estimate. Could you provide an adjusted estimate taking into account performance retribution and seniority of medical professionals?

Third, cost estimates for diagnostic procedures should be more accurately discussed. Particularly in early stages of the pandemic, diagnostic options were scarcely available and often relied on (at least partially) "in house" kit where available rather than on commercial ones (e.g. the Pediatric Hospital Bambino Gesù in Rome has published its specific experience on this topic). Therefore, some further explanation of the costs should be provided. Similar considerations could be moved for the costs of hospital stays, as during the early stages of the pandemic the reduced availability of PPE the costs of management skyrocketed in the whole of Italy, and the consequences of these costs have affected the management of Italian Hospital until today.

Formal issues:

- tables included in methods section (2-7) could be moved to annex section improving the overall readability of the paper

- Figure 5 is not properly formatted 

Ethical Issues

Even though I agree that the kind of data handling and reporting from this study does not elicit any ethical issue, Authors should provide some explanation about they guaranteed that data were correctly anonymized, handled and managed in order to cope with the GDPR requirements. The statement provided in later section is insufficient to clear potential claims about GDPR requirements (see point 26 of 679/2016 GDPR) Moreover, they could explain whether they had received or not (in this case, explain why this authorization was not needed) the authorization for using reported data from their parent organization.

Finally:

1) please revise the paper in order to be consistent with the usual format of Vaccine papers;

2) because of the main topic treated by Vaccines, the present paper should focus in more extensive details how vaccines have provided a potential sparing in total costs of SARS-CoV-2 pandemic management.

Comments on the Quality of English Language

The overall quality of the English text is quite appropriate.

Reviewer 2 Report

Comments and Suggestions for Authors

In this manuscript, the authors studied a very interesting topics, which is the economic burden of Covid-19 on the workforce of a teaching hospital in Central Italy. Now, during the post-pandemic era, it is very important to evaluate what happened during the pandemic, so that we could learn the lessons. The data collection and method part are clearly presented in this study.  However, the manuscript still has some flaws, and the authors need to carefully read through the manuscript and further modify the manuscript prior to publication.

1. In this manuscript, the format needs to be further modified. For example, line 79-80, 121-124, 159-167 and etc.

2. There are several 1.1.1 labeling in the Introduction part.

3. In the Figure 1, there are some non-English characters for “month”, which need to be modified.

4. Figure 5 is out of the page.

Reviewer 3 Report

Comments and Suggestions for Authors

The study was conducted in one of the teaching hospitals in Italy with intention to describe economic burden of Covid-19 costs in two research periods (March 1st, 2020 - February 9th, 2021, compared to February 10th, 2021 - March 31st, 2022). The costs of hospital workforce of hourly salaries lost during the illness (cost of productivity loss) and medical expenses e.g. cost of hospital admission, cost of at-home treatments and cost of contact tracing and screening tests were computed for both study periods.

The study supports the already known fact that the course of Covid-19 changed in severity due to change in virus virulence and protection gained from previous infections and vaccinations.

The present paper is clearly written and findings are of interest for experts in health economics and for medical community in general.

The paper is rather long – I suggest that Table 2, Table 3, Table 6, Figure 2, Figure 3 and Figure 4 are moved from the main text to supplementary material.

Round 2

Reviewer 1 Report

Comments and Suggestions for Authors

Estimated Authors,

even though I understand your comments, please take into account my previous comments:

1) In my previous review round, I did ask Authors to provide some insights about the background settings of the pandemic at the time of the survey: this issue remains substantially unsolved, as the only editing provided by Authors is "The study was carried out in a 1200 bed hospital in Rome": Authors should at least provide some further information - for example: was the Hospital a teaching one? did include an Emergency Department? What about the ICU size (number of beds) at the beginning of the pandemic and during the pandemic?

Moreover, I'm quite unconfortable with the answer provided by Authors regarding the aforementioned point: 

"We do not think that adding some epidemiological details on the occurrence of SARS-Co-2 pandemic could be an added value, since the aim of this study was to estimate the burden of disease and the related costs on the HW, before and after the beginning of the vaccination campaign for Covid-19 in Italy".

As the costs faced by Italian Healthcare System before and after the beginning of vaccination campaign for COVID-19 were mostly due to the management of COVID-19, even basic information about the pandemic at the time of study (therefore: not during the "first wave" but at least since December 2020 onwards) would improve the overall quality of the paper.

2) Authors have acknowledged that they were unable to check the seniority of involved healthcare professionals ("we have no possibility for checking for the seniority of healthcare professionals") Even though "this issue has been recognized as a possible limitation in the Discussion section", it was not truly discussed ("... we need to recognize that we were not able to retrieve performance retribution and seniority of healthcare professionals"). As previously stressed, while performance retribution for nurses and allied medical professionals is usually considered a marginal one, it could represent a significant share of the total retribution of medical doctors. As a large share of potential readers of this paper may be not familiar with Italian salaries, it should be more properly discussed and the subsequent limits in overall analyses more extensively discussed and acknowledged. Otherwise, it would appear quite strange that (as previously pointed out in my previous review) "in fact, the AVERAGE monthly income is lower than that of group DS".

3) My third comment was: "Third, cost estimates for diagnostic procedures should be more accurately discussed. Particularly in early stages of the pandemic, diagnostic options were scarcely available and often relied on (at least partially) "in house" kit where available rather than on commercial ones (e.g. the Pediatric Hospital Bambino Gesù in Rome has published its specific experience on this topic). Therefore, some further explanation of the costs should be provided. Similar considerations could be moved for the costs of hospital stays, as during the early stages of the pandemic the reduced availability of PPE the costs of management skyrocketed in the whole of Italy, and the consequences of these costs have affected the management of Italian Hospital until today". 

Your answer is quite interesting ("this was not the case for this hospital, since we acted as HUB for the performance of diagnostic testing of SARS-CoV-2 in Rome") as it provides further information about the characterization of your hospital, and should be provided when reporting your institution in introductory sections, but the topic remains not treated. Similar considerations about the costs of PPE, etc.

4) "The cost for hospital stay has been not changed, since this is fixed as value at the national level due to DRGs".

Thank you, but I guess we're facing a substantial misunderstanding. You're discussing on the payment by regional health authority for hospital stay, that in fact is based on DRG. My comment was about the REAL costs of hospital stays during the assessed time period. When unable, please discuss that while the reimboursment provided by health authority was pre-existing pandemic, the TRUE costs of hospital stays skyrocketed, contributing to the economic constraints of Italian healthcare system after the pandemic. The sentence "The Diagnosis Related Group (DRG) for Covid-19 foresees an average reimbursement of € 9,062 per patient requiring hospitalisation because of Covid-19" would then convey a far more significant message to the readers when they are aware that Italian NHS was forced to anticipate expenses that were only partially covered at Regional and National Level.

Moreover, as for previous comments: "All these issues were included in the Discussion section" would require a more extensive discussion of this topic.

5) Regarding the Ethical statement, I think that it is (now) quite appropriate. I would suggest to include the full reference from point 26 of 679/2016 GDPR reporting verbatim some sentences (i.e. "The principles of data protection should therefore not apply to anonymous information, namely information which does not relate to an identified or identifiable natural person or to personal data rendered anonymous in such a manner that the data subject is not or no longer identifiable. This Regulation does not therefore concern the processing of such anonymous information, including for statistical or research purposes.") Please keep in mind that other National Legal framework would impair the use of data that is reported in this study, therefore potential readers should be made aware that Italian National Law allows some degree of use of such information after appropriate anonymization, at least for scientific reasons.

Round 3

Reviewer 1 Report

Comments and Suggestions for Authors

Estimated Authors,

according to your reply, it is quite clear that Authors are not willingly to take in account the comments of the present reviewer. They have explained, in the rebuttal letter, the reasons underlying this choice, and I'm not quarreling about it. As my comments remain unsolved, and the underlying positive assessment of your study as well, I'm formally endorsing the acceptance but I'm forced to stress again my doubts.